# Mapping Cell-in-Cell Structures in Oral Squamous Cell Carcinoma

**DOI:** 10.3390/cells12192418

**Published:** 2023-10-08

**Authors:** Leonardo de Oliveira Siquara da Rocha, Bruno Solano de Freitas Souza, Ricardo Della Coletta, Daniel W. Lambert, Clarissa A. Gurgel Rocha

**Affiliations:** 1Gonçalo Moniz Institute, Oswaldo Cruz Foundation (IGM-FIOCRUZ/BA), Salvador 40296-710, BA, Brazil; leonardo.oliveira@ufba.br (L.d.O.S.d.R.); bruno.solano@fiocruz.br (B.S.d.F.S.); 2Department of Pathology and Forensic Medicine, School of Medicine, Federal University of Bahia, Salvador 40110-100, BA, Brazil; 3D’Or Institute for Research and Education (IDOR), Salvador 41253-190, BA, Brazil; 4Department of Oral Diagnosis, School of Dentistry, University of Campinas, Piracicaba 13414-903, SP, Brazil; 5Graduate Program in Oral Biology, School of Dentistry, University of Campinas, Piracicaba 13414-903, SP, Brazil; 6School of Clinical Dentistry, The University of Sheffield, Sheffield S10 2TA, UK; 7Department of Propaedeutics, School of Dentistry, Federal University of Bahia, Salvador 40110-150, BA, Brazil

**Keywords:** tumor spheroids, oral squamous cell carcinoma, cell cannibalism, cell-in-cell structures, cancer-associated fibroblasts

## Abstract

Cell-in-cell (CIC) structures contribute to tumor aggressiveness and poor prognosis in oral squamous cell carcinoma (OSCC). In vitro 3D models may contribute to the understanding of the underlying molecular mechanisms of these events. We employed a spheroid model to study the CIC structures in OSCC. Spheroids were obtained from OSCC (HSC3) and cancer-associated fibroblast (CAF) lines using the Nanoshuttle-PL^TM^ bioprinting system (Greiner Bio-One). Spheroid form, size, and reproducibility were evaluated over time (Evos^TM^ XL; ImageJ version 1.8). Slides were assembled, stained (hematoxylin and eosin), and scanned (Axio Imager Z2/VSLIDE) using the OlyVIA System (Olympus Life Science) and ImageJ software (NIH) for cellular morphology and tumor zone formation (hypoxia and/or proliferative zones) analysis. CIC occurrence, complexity, and morphology were assessed considering the spheroid regions. Well-formed spheroids were observed within 6 h of incubation, showing the morphological aspects of the tumor microenvironment, such as hypoxic (core) and proliferative zone (periphery) formation. CIC structures were found in both homotypic and heterotypic groups, predominantly in the proliferative zone of the mixed HSC3/CAF spheroids. “Complex cannibalism” events were also noted. These results showcase the potential of this model in further studies on CIC morphology, formation, and relationship with tumor prognosis.

## 1. Introduction

Over 350,000 new oral cancer cases are diagnosed yearly worldwide [1]. Oral squamous cell carcinoma (OSCC) represents over 90% of these [2,3,4], with a mortality rate of over 60% in five years [5,6]. The aggressive treatment regimens frequently required have severe impacts on the quality of life of survivors. Advances in the knowledge of OSCC pathogenesis, diagnostics, and therapeutics are hampered by the limited in vitro reproduction of the complex characteristics of tumor biology [7,8,9].

Among the events that take place in tumors, the occurrence of cell-in-cell (CIC) structures has drawn increasing attention because of their relationship with poorer prognosis in cancers such as lung cancer [10], breast cancer [11], and OSCC [12,13]. Classically known as “signet ring” or “bird’s eye” cells, these events are commonly defined as morphological findings of one cell within another [14,15,16,17], resulting from the distinct mechanisms of non-professional engulfment of living cells by their pairs [18]. Studies investigating the presence of CIC structures suggest that their incidence is a tumoral response to hostile circumstances, such as starvation [19,20], hypoxia [21], or chemo- and radiotherapy [11,22]. This mechanism enhances cell survival [23], immune escape [24], treatment resistance [25], and the selection of an aggressive cell population [26,27].

In addition, the tumor microenvironment (TME), composed mainly of non-tumoral cells, extracellular matrix, and blood vessels that support the tumor [28,29], also plays an essential role in tumor aggressiveness [30]. Cancer-associated fibroblasts (CAF), for example, interact with the tumor and the extracellular matrix, changing tumor metabolism [31,32] and motility [33], contributing to tumor invasion [34], resistance [35,36], and metastasis [37,38].

Commonly used two-dimensional cell culture models do not adequately reproduce many features of the tumor microenvironment, such as zone formation (e.g., quiescent and proliferative zones), oxygen and nutrient gradients, and complex cellular interactions [39,40]. Alternatively, preclinical research relies on animal models, such as mice or rats, which can characterize tumor pathogenesis [41]. Nevertheless, these methods have limitations, such as the inadequate reproduction of the systemic anti-tumoral immune response in immunodeficient animals used for tumor xenografts [41,42]. In this context, three-dimensional models aim to bridge the gap between in vitro cultures and in vivo tumors [29,43], simulating, for example, hypoxia and metabolic changes in the tumor microenvironment [44,45,46]. These models have been applied to study cancer complexities in the breast [47,48], kidney [49] carcinomas, neuroblastoma [50], and OSCC [51], for example.

Given the evidence of the contribution of CIC structures to the aggressiveness of OSCC, here we aimed to detail morphological aspects of these events, considering homotypic and heterotypic (CAF) cellular interactions and the formation of tumor zones, by employing a 3D bioprinting in vitro model to study these structures for the first time.

## 2. Materials and Methods

### 2.1. Cell Culture

The cell lines used in this study were a metastatic OSCC line obtained from the tongue (HSC3—JCRB, Osaka, Japan) and a primary cancer-associated fibroblast isolated from tongue OSCC (CAF1 [52], ethics committee approval number 4,706,681). Tumor cells were cultured in DMEM High-Glucose medium (Gibco^TM^, Waltham, MA, USA), supplemented with 10% fetal bovine serum (FBS—Gibco^TM^, Waltham, MA, USA) and 0.8% hydrocortisone (Sigma-Aldrich^TM^, Gillingham, UK). Fibroblasts were cultured in DMEM/F-12 medium (Gibco^TM^, Waltham, MA, USA), supplemented with 10% newborn calf serum (CALF—Gibco^TM^, Waltham, MA, USA). Both cultures were supplemented with 1% antibiotics (Pen Strep, Gibco^TM^, Waltham, MA, USA) and assessed for mycoplasma monthly. Cells were kept in incubators at 5% CO_2_ and 37 °C. A total of 70–80% of cell confluence was set for trypsinization (trypsin 0.5% 10×, Gibco^TM^, Waltham, MA, USA). CAFs were cultured up to the 10th passage.

### 2.2. Spheroid Formation

Tumor spheroids were obtained using the magnetic bioprinting Bio-Assembler^TM^ System (Greiner Bio-One, Kremsmünster, Austria). Briefly, this method relies on cellular membrane magnetization using biocompatible nanoparticles (Nanoshuttle-PL^TM^, Greiner Bio-One, Kremsmünster, Austria) composed of gold, iron oxide, and poly-L-lysine. Magnetic nanoparticles measured approximately 50 nm in diameter [53] and adhered electrostatically to the cellular membrane [54] for up to 8 days [55]. Their overall charge was a sum of the individual charges contributed by each component; nevertheless, they were biocompatible and did not interfere with cell functions, such as cellular viability, differentiation, proliferation, and phenotype, nor did they cause inflammation or oxidative stress [54,56]. Using a magnetic drive (30 pN/cell) under a 24-well repellent culture plate (Greiner Bio-One, Kremsmünster, Austria) for up to 24 h, cells were plated with Nanoshuttle-PL^TM^ aggregate in the form of non-adherent spheroids that hold together even after magnet removal.

Briefly, adhered cells were washed with PBS, dissociated, centrifuged, and counted (cell viability was evaluated by Trypan Blue staining using a Neubauer chamber). Cells were suspended in DMEM/F-12 Glutamax (Gibco^TM^, Waltham, MA, USA) completed with 10% fetal bovine serum (FBS—Gibco^TM^, Waltham, USA), 0.8% hydrocortisone (Sigma-Aldrich^TM^, Gillingham, UK), and 1% antibiotics (Pen Strep, Gibco^TM^, Waltham, MA, USA), and mixed with magnetic beads considering a standard ratio of 0.4 μL of Nanoshuttle-PL^TM^ for every 10,000 cells (4 × 10^−5^ nanoparticles/cell). To promote the interaction between nanoparticles and cells, three centrifugation steps (1500 rpm for three minutes each) were performed, after which the resulting suspension was plated in a 24-well repellent plate for bioprinting for 24 h using the underlying magnetic drive, kept in an incubator at 5% of CO_2_ and 37 °C. All experiments were conducted in triplicate. Figure 1 illustrates the steps of the spheroid formation method.

### 2.3. Imaging and Analysis Parameters

Spheroids were plated in two groups (homotypic HSC3-only and heterotypic HSC3/CAF spheroids) at two cell densities (1 × 10^3^ and 3 × 10^3^ cells per well). Heterotypic spheroids were also cultured in two cell-type ratios (2:1 and 1:1 HSC3:CAF). All spheroids were monitored and analyzed at 6 h, 12 h, 48 h, and 72 h after plating. At every time point, cells were imaged using an Evos^TM^ XL microscope (Thermo Scientific, Waltham, MA, USA) at 4× and 10× amplifications. The appraised parameters during imaging were spheroid size, evaluated by the largest diameter measured using ImageJ software version 1.8 (NIH, Bethesda, Rockville, MD, USA); spheroid form (roundness and cell density); distribution of cells between spheroid center and margin; and variability among replicas.

### 2.4. Histology and CIC Count

At 6 h, 12 h, 24 h, 48 h, and 72 h, two spheroids were collected from each experiment for histological processing. Briefly, plating media was removed from the well, and spheroids were collected using previously cut 1000 μL tips to avoid fragmentation. Each spheroid was kept in cold formaldehyde at 4% for at least 24 h, followed by paraffin embedding and sectioning into 4 μm slices. Sections were mounted onto slides and scanned using an Axio Imager Z2/VSLIDE Scanner (Zeiss Microscopy, Oberkochen, Germany). Analyses were made using the OlyVIA visualizer version 2.4 (Olympus Lifescience, Tokyo, Japan). In addition, Appendix A shows the histological features of oral squamous cell carcinoma tissue.

For histomorphological analysis, the following aspects were considered [57,58] (Figure 2): tumoral zones (central/necrotic and marginal/proliferative), pleomorphism, and atypical cellular findings, visualization and location of CIC structures, identified by the morphological definition of “bird’s eye” or “signet ring” cells [15,59], i.e., outer cell well or partially outlined with a semilunar nucleus pushed towards the cellular periphery [60,61].

Semiquantitative analysis was performed in sections obtained from 40× amplified scanned slides. To count the CIC structures, one section from each of the two spheroids was selected from each group (homotypical and heterotypical) at each time point. For each sample, areas of 181 μm × 97 μm were chosen randomly from each zone: one area from the spheroid margin (periphery) and one from the spheroid inner core (central). Results were expressed as a percentage corresponding to the number of CIC structures/total number of well-defined cells in the area. Analyses were performed using ImageJ software version 1.8 (NIH, Bethesda, USA).

### 2.5. Statistical Analysis

Results were analyzed using GraphPad Prism version 8 (Dotmatics, Boston, MA, USA). Data were evaluated for normality and Gaussian distribution, with parametric and nonparametric tests applied accordingly (Mann–Whitney test and Student’s *t*-Test). When determining differences between the two independent groups by comparing three or more independent groups, the *p*-value was considered less or equal to 5% for sampling variability.

## 3. Results

### 3.1. Spheroid Model Establishment

After plating, spheroids were monitored after 6 h, 12 h, 24 h, 48 h, and 72 h, while bioprinting was performed for 24 h (Figure 3A). Round-shaped spheroids with dense cellular aggregation at their center were seen from 6 h (Figure 3B). At 6 h, 12 h, and 24 h, heterotypic spheroids had a significantly larger diameter (*p* = 0.025), while a trend of time-dependent reduction in spheroid size (contraction) under all conditions was noted. This only reached statistical significance in the heterotypic spheroid group at 12 h (Figure 4). High reproducibility was also observed between the samples.

### 3.2. Histomorphological Assessment

Histomorphological analysis using hematoxylin and eosin staining showed a spheroid morphology compatible with the formation of OSCC tumor zones, which can be further observed in the samples of OSCC tissues (Appendix A). Tumor cells exhibited high pleomorphic and polyhedral aspects, consistent with the OSCC pattern (Figure 5A), whereas at the periphery of the spheroid (outer boundary), cells were cohesive and had a flatter shape (Figure 5B). In the heterotypic spheroids, we observed fusiform and elongated cells with a fibroblast-like morphology (Figure 5C). A necrotic center was observed in all groups, characterized by an amorphous area of eosinophilic staining and apoptotic cells. In homotypic spheroids, the necrotic center was predominantly singular, whereas in heterotypic groups, the presence of multiple sparse necrotic areas was noted (Figure 6A). Cellular pleomorphism and nuclear hyperchromatism were more evident in the proliferative zone (at the periphery of the spheroid). Furthermore, we observed areas of cellular loosening, starting from the edge of the spheroid (Figure 6B).

### 3.3. Identification of Cell-in-Cell Events

To assess the presence of CIC structures, the morphological description of “signet rings” or “bird’s eye cells” was defined as a parameter for the identification of these events. CIC structures were observed in all regions of both homotypic and heterotypic spheroids. In most cases, the outer cell’s nucleus presented a classic “spindle-shaped” form, whereas the inner cell’s nucleus was not always visible. Meanwhile, the cellular membranes of the involved cells did not always present visible continuity. These findings varied in size, roundness, integrity of the involved cells, and proximity to one another, sometimes appearing as clusters of CIC events (Figure 7). CIC structures were more abundant in the spheroids of HSC3 cells cultured with CAFs (Kruskal–Wallis, *p* = 0.0037 and 0.0464) (Figure 8A) and in proliferative areas (ANOVA, *p* = 0.0062), closer to the margin of the spheroids (Figure 8B). In some cases, engulfed cells were found within other internalized cells, a phenomenon that has been described as “complex cannibalism” [20] (Figure 9).

## 4. Discussion

Since the first reports of CIC findings were made over 100 years ago [17,62,63], they have become common findings in malignancies such as lung cancer [64], breast cancer [11,65], and oral cancer [12,66], and have been linked to an increase in tumor aggressiveness [67,68], resistance [22] and survival [69,70,71], leading to worse prognosis [25,72]. However, despite being easily identifiable in routine H/E stains and electron microscopy [69,73,74,75,76], there is a need for models that enable further study of these events in vitro. Herein, we demonstrate that CIC structures can be studied in 3D bioprinted spheroids, which reproduce histomorphological features of the OSCC.

Despite bioprinting spheroids having been used to study various cancer types such as glioblastoma [53,77], breast cancer [78], pancreatic duct adenocarcinoma [55], osteosarcoma [79] and ovarian cancer [80], there is a lack of studies that validate this model for OSCC. The spheroid bioprinting method might represent an advantageous approach compared to hanging drop cultures [81], organ-on-a-chip [82], and organoids [83] for allowing higher reproducibility, technical simplicity, and the ability to construct heterotypic spheroids with well-nutrient distribution and tumor zone formation [84,85,86]. This approach cooperates in dealing with the lack of appropriate methods to study CIC structures [11,18,87]. Meanwhile, the reproduction of heterotypic cell–cell interactions, heterogenous distribution of oxygen, and consequently the formation of tumor zones [55,88,89], all of which influence the formation of CIC structures [18,30], confer an advantage to the use of spheroid models in this field.

Despite no correlation between CIC frequency and time of observation, our findings show that CIC structures were more frequent in the proliferative zone of heterotypic OSCC/CAF spheroids. The occurrence of heterotypical interactions involving mesenchymal and cancer cells in spheroids has been described in breast cancer [11,90] and pancreatic cancer [85], but not to the best of our knowledge in OSCC. CIC findings have been associated with aggressiveness hallmarks, such as cell invasion and resistance [26,91], as well as stemness [92,93]. In addition, it is well established that CAFs play an essential role in tumor metabolism [31,32], invasion and metastasis [94,95,96], autophagy [97], and therapeutic resistance [98]. Further immunophenotyping studies using spheroid sections may be helpful in matching CAF and CIC identification with the expression of markers such as NANOG and SOX2, considering their confluent association with tumor prognosis [11,92,93]. Additionally, given the production of the extracellular matrix by HSC3 cells [99] and the role of CAFs in remodeling such matrix [31,32], it is important that further studies investigate the role of cellular–matrix interactions in the formation of CIC structures.

Despite cellular interactions being ubiquitous [100], in our results, CIC structures appeared more often in the proliferative zone. It is important to emphasize that within this tumor microenvironment model, cancer cells exhibit enhanced plasticity and an invasive phenotype [57,101,102]. Meanwhile, previous studies show that the fibroblasts are predominantly located in the periphery of the spheroids [103,104], which could further suggest the participation of these cells in CIC formation. In contrast, some studies indicate hypoxia as a relevant aspect for triggering the formation of CIC [21,26] due to its relationship with tumor metabolism [105,106]. Nevertheless, it is important to acknowledge that the conventional histomorphological characterization of CIC structures alone is inadequate in identifying the cell types involved in their formation, as well as if the involved cells are undergoing cell death [24,60,107], which could explain less detection of CIC features in necrotic zones, despite also having been observed in this study. Immunophenotyping assays can aid in cellular identification and in understanding the mechanisms behind the located CIC events (e.g., cannibalism and entosis).

In summary, in this study, we employed a spheroid bioprinting method to study CIC events in oral cancer for the first time. CIC was detected throughout the spheroids but more commonly in the proliferative area of spheroids in which cancer cells were cocultured with CAFs. This study demonstrates that the spheroid bioprinting model described here offers a reliable and practical approach for studying the mechanisms underpinning the formation of CIC structures; this may lead to the identification of prognosis biomarkers and possible therapeutic targets in translational oncology.

## Figures and Tables

**Figure 1 cells-12-02418-f001:**
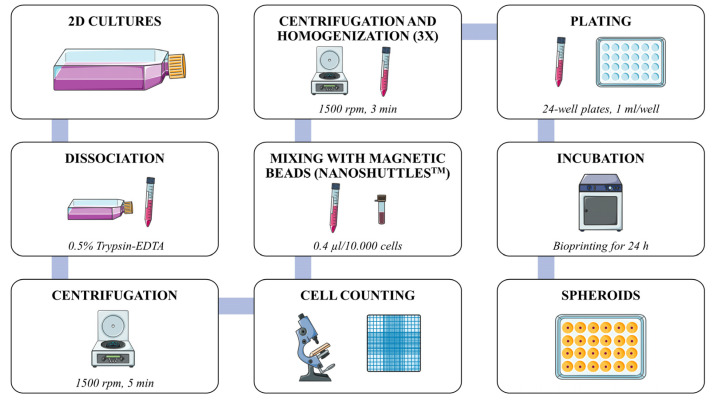
Flowchart of 3D bioprinted spheroid formation framework. Created using Smart Servier images (Creative Commons Attribution 3.0 France). Nanoshuttle-PL^TM^ is a trademark of Greiner Bio-One.

**Figure 2 cells-12-02418-f002:**
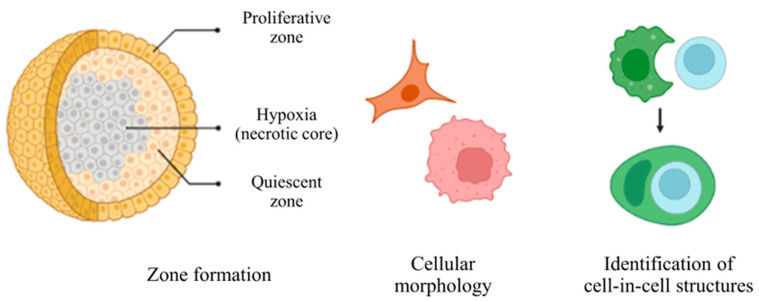
Representation of histomorphological features observed during H&E analysis. Created using Biorender.com (accessed on 30 April 2023).

**Figure 3 cells-12-02418-f003:**
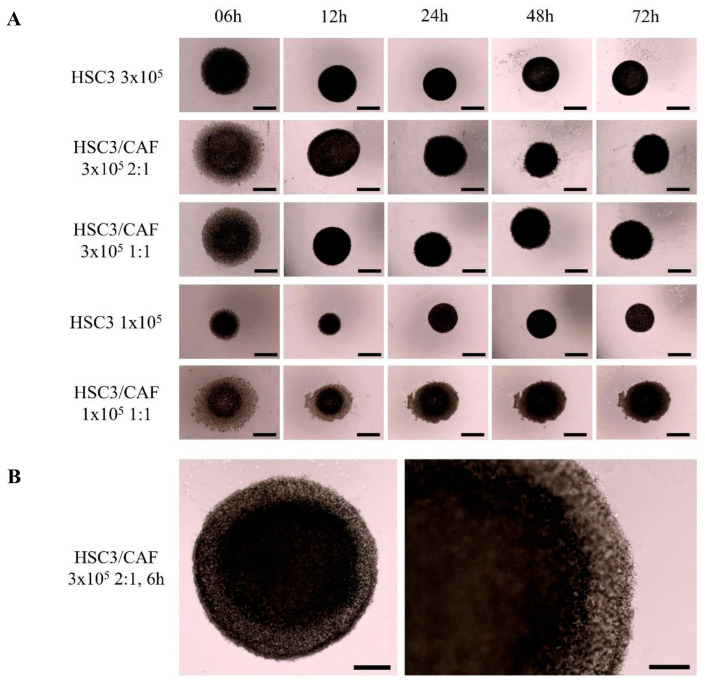
Oral squamous cell carcinoma (HSC3 line) homotypic and heterotypic (co-cultured with cancer-associated fibroblasts, CAF) spheroids. (**A**) Representative images of spheroids obtained with different cell plating densities, cell line combinations, and ratios (lines), taken at different timepoints (from the start of the 24 h bioprinting) (columns). At 6 h, spheroid formation can already be seen, reaching appropriate cell aggregation after 24 h. Images obtained with Evos^TM^ XL. Scale bar: 1000 μm. (**B**) Representative images of cell organization. A central region with higher cell density can be seen, surrounded by an outer layer of sparse cellular organization, typical of a proliferative zone. Images obtained using Evos^TM^ XL. Scale bar: 500 μm (left) and 250 μm (right).

**Figure 4 cells-12-02418-f004:**
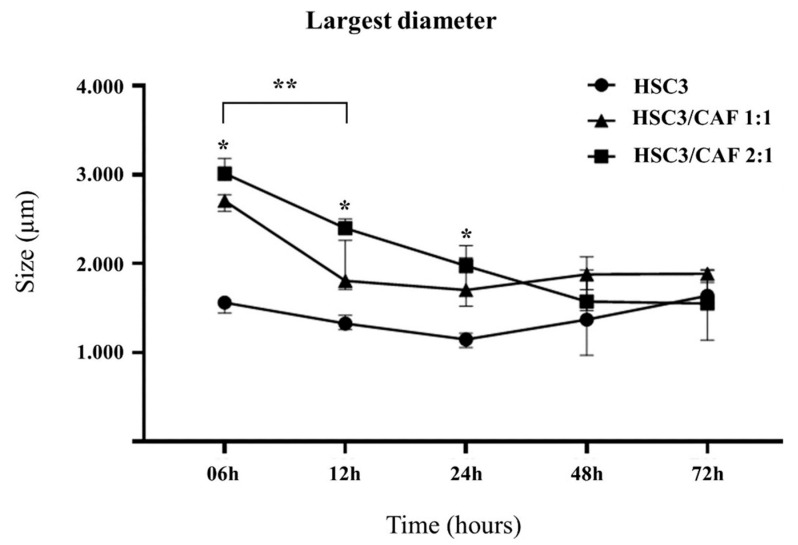
Median and standard deviation of spheroid diameter of homotypic and heterotypic groups (co-cultured with cancer-associated fibroblasts, CAF) in the evaluated timepoints. * Spheroid size in the heterotypic group size was bigger than in the homotypic group at 6 h, 12 h, and 24 h. ** Spheroid contraction was larger at 12 h. Image made with GraphPad Prism version 8.

**Figure 5 cells-12-02418-f005:**
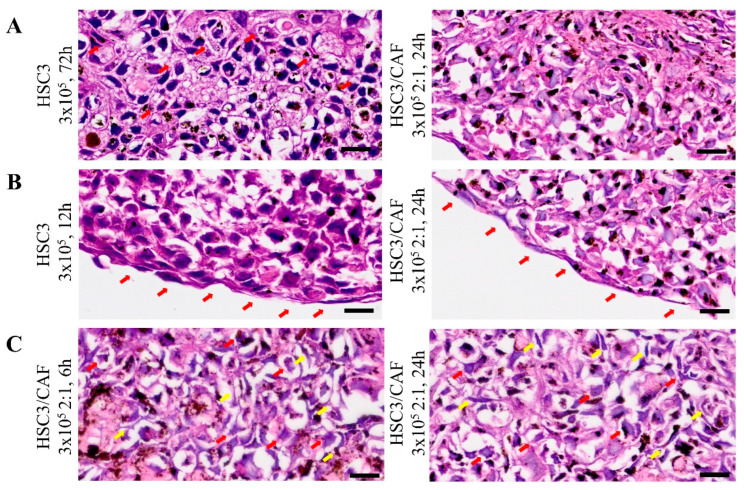
Histomorphological findings in 4 µm thickness sections obtained from oral squamous cell carcinoma (HSC3 line) spheroids, both homotypic and heterotypic (co-cultured with cancer-associated fibroblasts, CAF). (**A**) Pleomorphism seen in the tumor cell population and the formation of intercellular bridges can be seen (red arrows). (**B**) In the periphery of the spheroid, cells with a flatter morphology can be observed (red arrows). (**C**) Heterogeneous grouping of polyhedral cells (red arrows) and fusiform and elongated cells (yellow arrows). Images obtained using Axio Imager Z2/VSLIDE. Coloring: H/E. Scale bar: 20 μm.

**Figure 6 cells-12-02418-f006:**
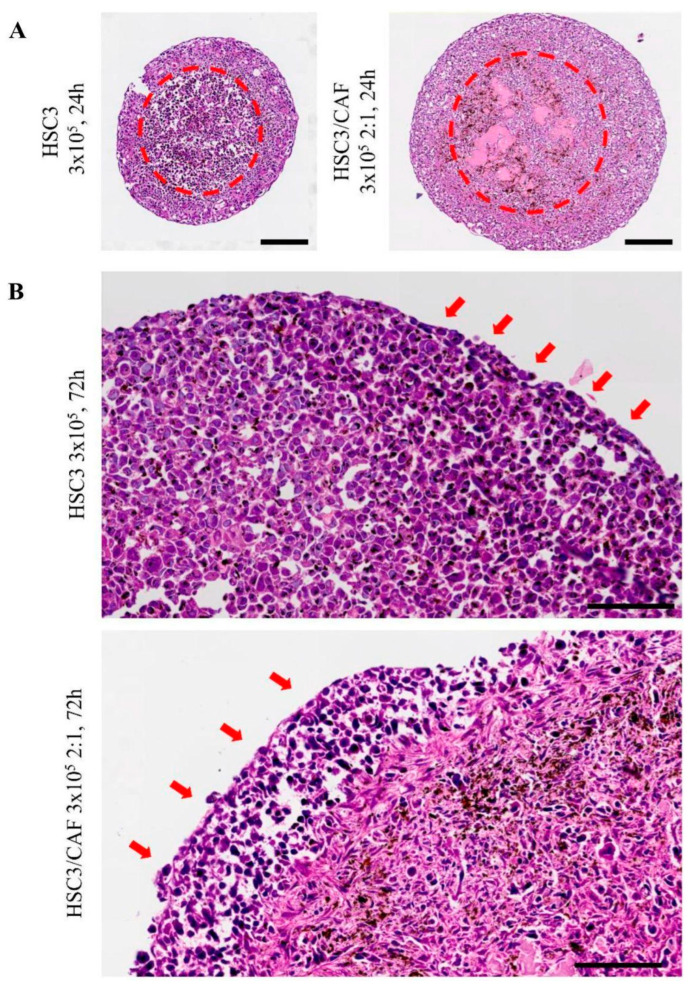
Histomorphological aspects in 4 µm thickness sections obtained from oral squamous cell carcinoma (HSC3 line) spheroids, both homotypic and heterotypic (co-cultured with cancer-associated fibroblasts, CAF). (**A**) Necrotic cores (dotted red line) formed in homotypic (**left**) and heterotypic (**right**) spheroids. In the homotypical groups, the loss of cellular adhesion and tissue pattern suggest liquefactive necrosis. In mixed spheroids, multiple focal amorphic and eosinophilic areas were noticed, due to a difference in extracellular matrix deposition, and suggest a pattern of ischemic necrosis. Scale bar: 200 μm. (**B**) Towards the periphery of the spheroid, a pattern of loss of cellular adhesion and hyperchromatic nuclei was frequent (red arrows), representative of necrotic cells. Images obtained using Axio Imager Z2/VSLIDE. Coloring: H/E. Scale bar: 200 μm (**top**) and 100 μm (**bottom**).

**Figure 7 cells-12-02418-f007:**
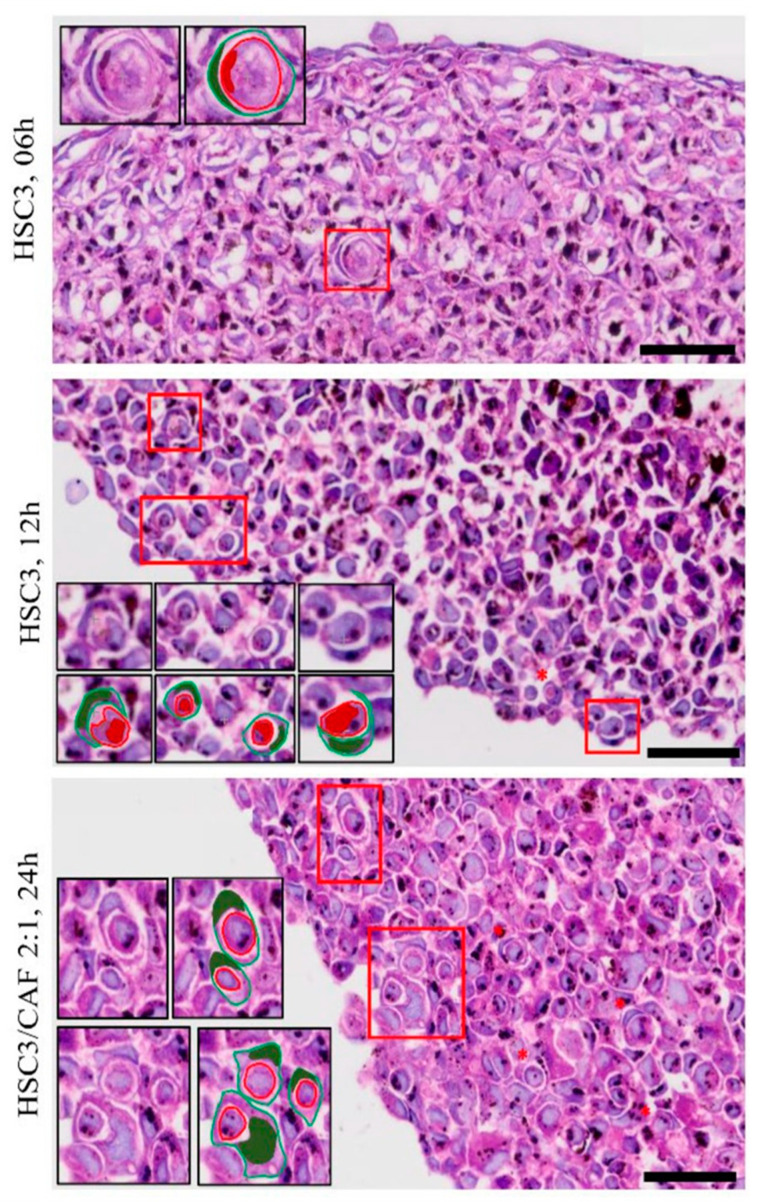
Cell-in-cell structures, described as “signet ring” or “bird’s eye” cells, found in 4 μm thickness sections of homotypical and heterotypical spheroids of oral squamous cell carcinoma co-cultured with cancer-associated fibroblasts. Red asterisks indicate cell-in-cell structures and, in detail (red box), graphical representations of cell-in-cell structures (external cell and its nucleus in green, and internal cell with or without its nucleus shown in red). Image obtained using Axio Imager Z2/VSLIDE. Graphic details made by the author. Coloring: H/E. Scale bar: 20 μm. CAF = cancer-associated fibroblasts.

**Figure 8 cells-12-02418-f008:**
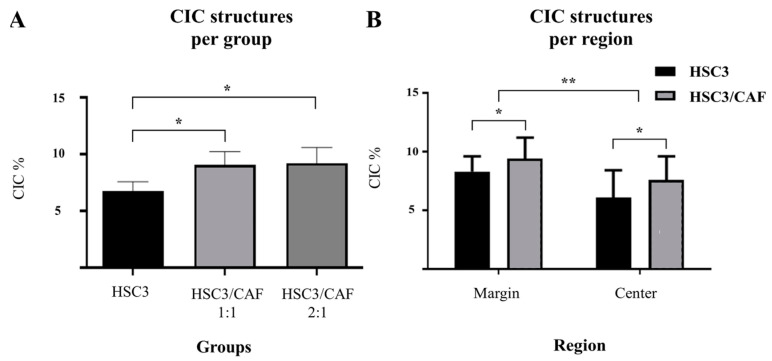
Data on cell-in-cell structures found in oral squamous cell carcinoma spheroids, both homotypical and heterotypical (co-cultured with cancer-associated fibroblasts, CAF), expressed in percentage of total cell-in-cell structures per total amount of cells detected within a predefined region. (**A**) Cell-in-cell (CIC) structures were more frequent in heterotypic spheroids (*). (**B**) Cell-in-cell structures were more frequent in the periphery (proliferative zone) and center (necrotic core) of heterotypic spheroids (*), and, in both groups (homotypic and heterotypic), more structures were found in the proliferative zone (**). Images made with GraphPad Prism version 8.

**Figure 9 cells-12-02418-f009:**
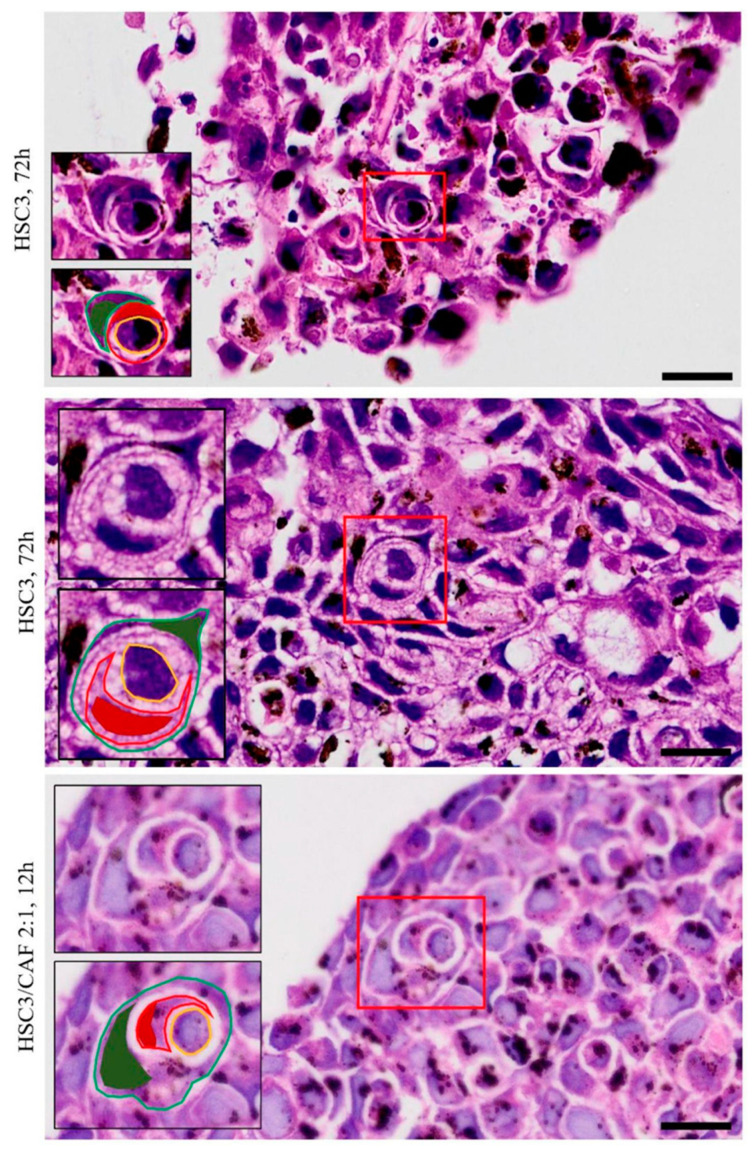
Cell-in-cell structures, described as “complex cannibalism”, in which a cell is found within a cell within another cell, found in 4 μm thickness sections of homotypical and heterotypical spheroids of oral squamous cell carcinoma co-cultured with cancer-associated fibroblasts. In detail (red box), the outer cell, and its nucleus (in green), the “middle” cell and its nucleus (in red), and the inner cell (in yellow). Image obtained using Axio Imager Z2/VSLIDE. Graphic details made by the author. Coloring: H/E. Scale bar: 20 μm. CAF = cancer-associated fibroblasts.

## Data Availability

The data presented in this study are available on request from the corresponding author.

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
