# Peer review of "Mapping Cell-in-Cell Structures in Oral Squamous Cell Carcinoma"

_cells, 2023, doi:10.3390/cells12192418_

Round 1

Reviewer 1 Report

Dear,

I read this manuscript and it was in my interest. Authors investigated a spheroid model to study CIC structures in OSCC.1/ Introduction needs to be improved and I suggest the following references: *Khayatan D, Hussain A, Tebyaniyan H. Exploring animal models in oral cancer research and clinical intervention: A critical review. Vet Med Sci. 2023. *Tahmasebi E, Alikhani M, Yazdanian A, Yazdanian M, Seifalian A. The current markers of cancer stem cell in oral cancers. Life Sci. 2020;249:117483. 2/Method and materials are well-explained.  3/Results are clear. 4/ Discussion is poor and it would be to use the above mentioned references.

Best,

Author Response

Response attached below.

Reviewer 2 Report

Dear authors, the comments are in attached file. 

Author Response

Response attached below. Thank you for your considerations.

Reviewer 3 Report

Cell-in-cell (CIC) structures contribute to tumor aggressiveness and poor prognosis in oral squamous cell carcinoma (OSCC). However, the article does not provide a detailed explanation of what CIC structures are or how they contribute to tumor aggressiveness. Study of CIC structures in OSCC using a spheroid model may provide new insights into the molecular mechanisms involved in tumor aggressiveness and help identify new therapeutic targets. This study is recommended for publication in the present form.

Author Response

Response attached below.

Round 2

Reviewer 1 Report

Dear,

It is acceptable.

Best,

Author Response

Dear reviewer, thank you for your careful considerations which have improved our paper.

Reviewer 2 Report

Dear authors, thank you for comprehensive response to my comments. I know that spheroids are "difficult" object for histology, and wish you success  in their studies. I think application of additional staining may be useful in identification of cell different types. 

I am satisfied by the responses.    

Minor editing could be useful, in total the text is readable and understandable 

Author Response

Dear reviewer, we thank you for taking the time to review our paper and for your insightful considerations.